# Splicing activates transcription from weak promoters upstream of alternative exons

Maritere Uriostegui-Arcos [1], Steven T. Mick [1], Zhuo Shi [2], Rufuto Rahman[1] & Ana Fiszbein [1] ✉

Transcription and splicing are intrinsically coupled. Alternative splicing of internal exons can fine-tune gene expression through a recently described phenomenon called exon-mediated activation of transcription starts (EMATS). However, the association of this phenomenon with human diseases remains unknown. Here, we develop a strategy to activate gene expression through EMATS and demonstrate its potential for treatment of genetic diseases caused by loss of expression of essential genes. We first identified a catalog of human EMATS genes and provide a list of their pathological variants. To test if EMATS can be used to activate gene expression, we constructed stable cell lines expressing a splicing reporter based on the alternative splicing of motor neuron 2 (SMN2) gene. Using small molecules and antisense oligonucleotides (ASOs) currently used for treatment of spinal muscular atrophy, we demonstrated that increase of inclusion of alternative exons can trigger an activation of gene expression up to 45-fold by enhancing transcription in EMATS-like genes. We observed the strongest effects in genes under the regulation of weak human promoters located proximal to highly included skipped exons.

RNA splicing is a highly regulated process that influences almost every aspect of eukaryotic cell biology. Early studies suggest that both spliceosome assembly and catalysis of splicing occur in a co-transcriptional manner[1–3], creating opportunities for functional connections between transcription and splicing. A key player mediating the effects of transcription on RNA splicing is the RNA polymerase II (RNAPII) itself[4], as post-translational modifications of its C-terminal domain (CTD) can create a binding platform for splicing factors (recruitment model), as well as affect the rate of transcription elongation that modulates downstream splicing decisions (kinetic model)[5–7]. Since chromatin structure can influence transcription rate and recruitment of factors, histone modifications are a powerful source of splicing regulation[8,9]. More recently, accumulating evidence suggests a "reverse-coupling" mechanism in which splicing feeds back into transcription[10,11]. Adding an intron to an intron-less gene often boosts different stages of gene expression in plants, fungi, and animals by a phenomenon known as intron-mediated enhancement of gene expression[11]. Moreover, a functional 5′-splice site in close proximity to

a promoter was found to enhance transcriptional output[10] and change chromatin structure[12]. We recently characterized a related phenomenon called Exon-Mediated Activation of Transcription Starts (EMATS), in which the splicing of internal exons impacts the spectrum of promoters used and the expression levels of the host genes[13]. We observed that EMATS modulates gene expression through changes in splicing efficiency of internal exons during evolution. The strongest effects are seen when there is a highly included (included in most transcripts, see Methods section) skipped exon (SE) located downstream and proximal to a weak promoter. We also showed that specific proteins involved in splicing, whose depletion have large effects on alternative promoter use, have widespread interactions with core transcription machinery and that the splicing factor HNRNPU recruits core transcription machinery locally. Overall, our observations support a reversed "recruitment model" in which specific splicing factors recruit core transcription machinery to the vicinity of transcripts as they are being transcribed, boosting RNAPII occupancy and activity of nearby promoters.

[1]Biology Department, Boston University, Boston 02215, USA. [2]Biology Department, Massachusetts Institute of Technology, Cambridge 02139, USA. ✉e-mail: anafisz@bu.edu

Dysregulation of splicing, spliceosome complexes, and RNA processing can lead to diseases including tauopathies, muscle disorders, hypercholesterolemia, and cancer[14,15]. A deep dive into understanding how RNA splicing contributes to gene expression programs is critical to uncovering pathophysiological mechanisms of diseases. In EMATS, splicing changes are associated with usage of alternative promoters with implications in gene expression programs in a tissue- and species-specific fashion. However, the role of EMATS genes in human diseases is still not characterized.

Small molecules that target specific DNA sequences have been widely used to control gene expression[16], and small-molecule drugs that target RNA have also emerged as a new opportunity to therapeutically modulate numerous cellular processes[17]. More recently, antisense oligonucleotides (ASOs) have emerged as a specific, rapid and potentially high-throughput approach for modulating gene expression through recognition of cellular RNAs[18]. While small molecules and ASOs have been successfully used to treat several diseases, most of these therapeutics have focused on gene silencing. One exception is the recent usage of ASOs to inhibit non-productive splicing products that introduce premature termination codons and are degraded by nonsense-mediated decay (NMD)[19]. Examples of successful, therapeutic small molecules and ASOs that target splicing are Risdiplam[20,21] and Spinraza[22], respectively, which are designed to upregulate splicing of exon 7 in the motor neuron 2 (SMN2) gene. Splicing rescue of exon 7 in SMN2 makes up for the loss-of-function of the paralogous gene, SMN1, which is mutated in spinal muscular atrophy (SMA) patients[22,23]. Recent studies show that chromatin modifiers cooperate with ASOs to control splicing and improve neuromuscular function in SMA[24].

Here, we present a catalog of EMATS genes and investigate their role in human genetic diseases. Using small molecules and ASOs, we developed a therapeutic strategy to use EMATS to activate gene expression. We demonstrated that increased exon inclusion of a splicing reporter based on SMN2 can trigger a 45-fold upregulation in gene expression through transcriptional activation and identify the conditions to apply this strategy for treatment of EMATS-associated genetic diseases.

## Results

### A catalog of human EMATS genes

Working with evolutionarily-new mouse exons, we previously observed that EMATS requires splicing of an internal exon, not merely presence of a 5′ or 3′ splice site, to activate a cryptic promoter, and the effect is stronger when the promoter is intrinsically weak and exon splicing is efficient[13]. EMATS is sensitive to the genomic distance between exon and upstream promoter pair, showing the strongest effects when there are 1–5 kilobases (kb) between them. Since these changes in gene expression have potential therapeutic benefits, here we aimed to identify human genes that have an EMATS structure, in which their transcription and translation activity could be modulated through changes in splicing.

Using the chromosomal GENCODE[25] annotation (release 26), we started with 19,817 protein coding genes (Fig. 1a). Based on the genome annotations, we collapsed all overlapping exons from the same gene into one "meta-exon" that accounts for alternative splice sites, alternative transcription start sites, and alternative polyadenylation sites. Exons were determined to be alternative or constitutive by examining substitute exon inclusions in different transcripts. We then used the Hybrid-Internal-Terminal (HIT index) method to classify exons based on their isoform-specific transcript usage[26]. Percent spliced-in (PSI) levels of inclusion for alternative exons were measured using the HIT index for first and last exons and rMATS[27] for internal exons across 54 tissue sub-types and hundreds of individuals, amounting to the complete list of over 17,000 samples from the GTEx project (Supplementary Fig. 1a). We defined weak alternative exons as those with median PSI values across tissues below the median value of all exons with the same classification, and strong alternative exons as those with median PSI values above the classification-wide median. To narrow our focus, we first identified human protein-coding genes that possessed at least one skipped exon (SE) and two alternative first exons (AFE). We then further refined our selection by choosing genes in which a weak AFE (wAFE) was located within 5 kb upstream of one or more strong SE's (sSE) 5′ coordinate (Fig. 1a). Our complete list of 2,306 human EMATS genes include 4,510 unique AFE/SE pairs (Supplementary Data 1).

When applying the EMATS criteria to individual tissues, we uncovered 687 tissue-specific EMATS genes, 219 of which are not represented in the inter-tissue gene set (Fig. 1b, Supplementary Fig. 1b, Supplementary Data 2). Consistent with their transcriptional diversity, we identified 85 testis- and 174 brain-specific EMATS genes, amounting to 3.7- and 7.6-fold more tissue-specific genes than the 22.9 average. As expected, the abundance or scarcity of tissue-specific EMATS genes anticipates how well-represented the tissue is in GTEx, as well as how a tissue's EMATS gene profile compares to other tissues. Brain tissue sub-types, for instance, preferentially cluster together, while testis and whole-blood tissues—both having high numbers of tissue-specific genes but lacking additional sub-types—do not participate in any cluster (Fig. 1c). Detailed code to identify EMATS genes is provided at https://github.com/fiszbein-lab/emats-genes.

We found that human genes with EMATS structure are longer genes (Fig. 1d) with shorter transcripts (Supplementary Fig. 1c) that produce significantly more alternative isoforms compared to non-EMATS genes (Supplementary Fig. 1d). Consistent with similar analyses in other species, human EMATS genes are enriched in transcription and translation regulatory activities (Fig. 1e). We recently classified a category of exons that we call hybrid exons which are used as terminal and internal exons in different transcripts[26]. We identified ~100,000 human inter-tissue hybrid exons which are used as terminal exons in one tissue but as internal exons in other tissues, and ~20,000 intra-tissue hybrid exons which are used as hybrid within the same tissues. Notably, we found that ~35% of AFEs in an EMATS structure are hybrid exons while the majority are obligate first exons (Fig. 1f). As expected, obligate first exons in EMATS genes are mostly located towards the 5′ end of genes, while hybrid first internal exons and skipped exons in an EMATS structure are distributed more uniformly across gene bodies (Fig. 1g). These findings reinforce the notion that EMATS is a mechanism that influences transcription starts with potentially crucial implications in gene regulation without altering protein coding structures.

### EMATS genes are associated with genetic diseases and changes in gene expression

In order to provide a framework to study the role of EMATS in human diseases, we assembled a master gene-phenotype dataset from Online Mendelian Inheritance in Man (https://omim.org/) to obtain a full list of genes involved in human genetic diseases[28,29]. OMIM is a continuously updated compendium of human genes, diseases, and traits with in-depth information on the molecular mechanisms and phenotypic expression. This dataset was merged with the EMATS structure dataset to obtain a list of EMATS genes with connections to human diseases (Fig. 2a). To find variants within these genes that commonly cause disease, we pulled variant data from ClinVar[30]. Variants with pathological or likely pathological annotations were selected and combined with the above data to identify variants affecting the exons in EMATS structure. This led to a list of 573 EMATS genes with thousands of variants falling in 1,334 regions essential to the EMATS structure (Fig. 2a). We identified genetic diseases involving exons with EMATS structure and found enrichment for intellectual development disorders, neurodevelopmental disorders, immunodeficiency, cancer, deafness, and others (Fig. 2a). We then explored the specific genomic

position of the genetic variants and divided them into upstream of the EMATS AFE/SE, in the TSS/3'splice site region, within the EMATS AFE/SE region, in the 5' splice site region, and downstream of the EMATS AFE/SE. We found that the majority of variants are located within the EMATS AFEs and SEs, with a significant number also located within the exons' splice sites (Fig. 2b).

We recently showed that during evolution, the gain of new exons is associated with increased gene expression levels[13]. Here, we wondered whether the splicing of skipped exons remains associated with host gene expression in human transcriptomes. We first analyzed the association between gene expression and alternative splicing events across over 17k human samples including 54 tissue sub-types using data from the GTEx project[31]. Although all splicing events are positively associated with gene expression, splicing of skipped exons showed the largest effect (Fig. 2c). Consistent with our findings during evolution, higher inclusion levels of skipped exons measured in SE PSI are associated with higher gene expression levels across human tissues on a global scale (Fig. 2d) and for individual genes (Fig. 2e, Supplementary

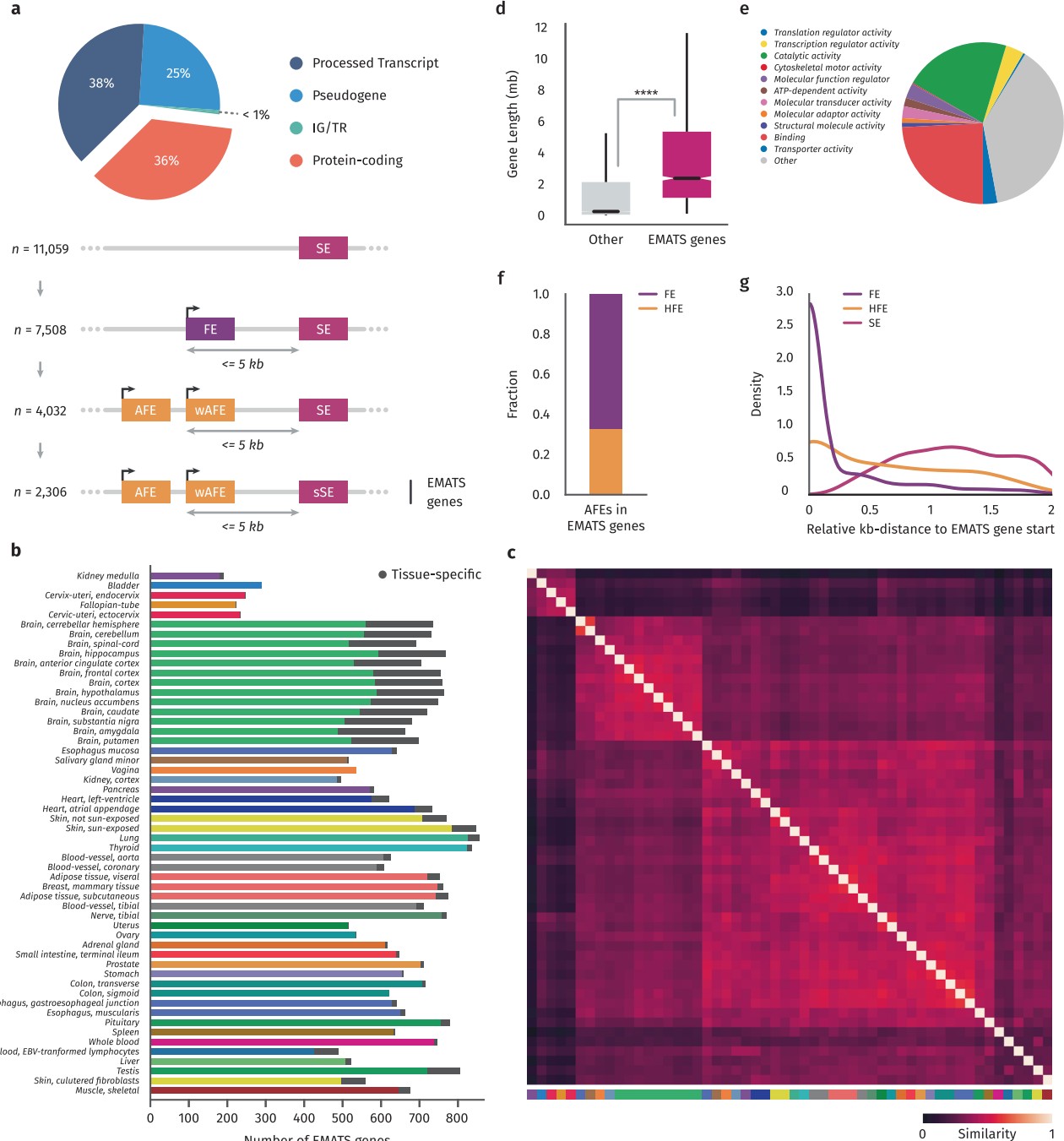

**Fig. 1 | Identification and characterization of human EMATS genes. a** A step-by-step identification of human EMATS genes with the following characteristics: protein coding genes, with multiple alternative promoters, and at least one weak AFE located within 5 kb upstream of a strong SE. **b** Number of EMATS genes identified in each human tissue and sub-tissue using data from the GTEx project. Colors represent tissues and grey bars represent tissue-specific EMATS genes, identified only in that tissue. **c** Heatmap showing the overlap between EMATS genes identified in each tissue and sub-tissue. **d** Distribution of total gene length for genes with EMATS structure and genes without EMATS structure. Statistical significance is indicated by asterisks (****$p < 0.0001$; two-sided independent t-test); $n$ (other genes) = 55,864, and $n$ (EMATS) = 2306. **e** Enrichment of human EMATS genes in gene ontology classifications. **f** Percentage of first exons in an EMATS structure classified as obligate first exons (FE) or first internal hybrid exons (HFE). **g** Relative positions of FE, HFE and skipped exons (SE) in an EMATS structure.

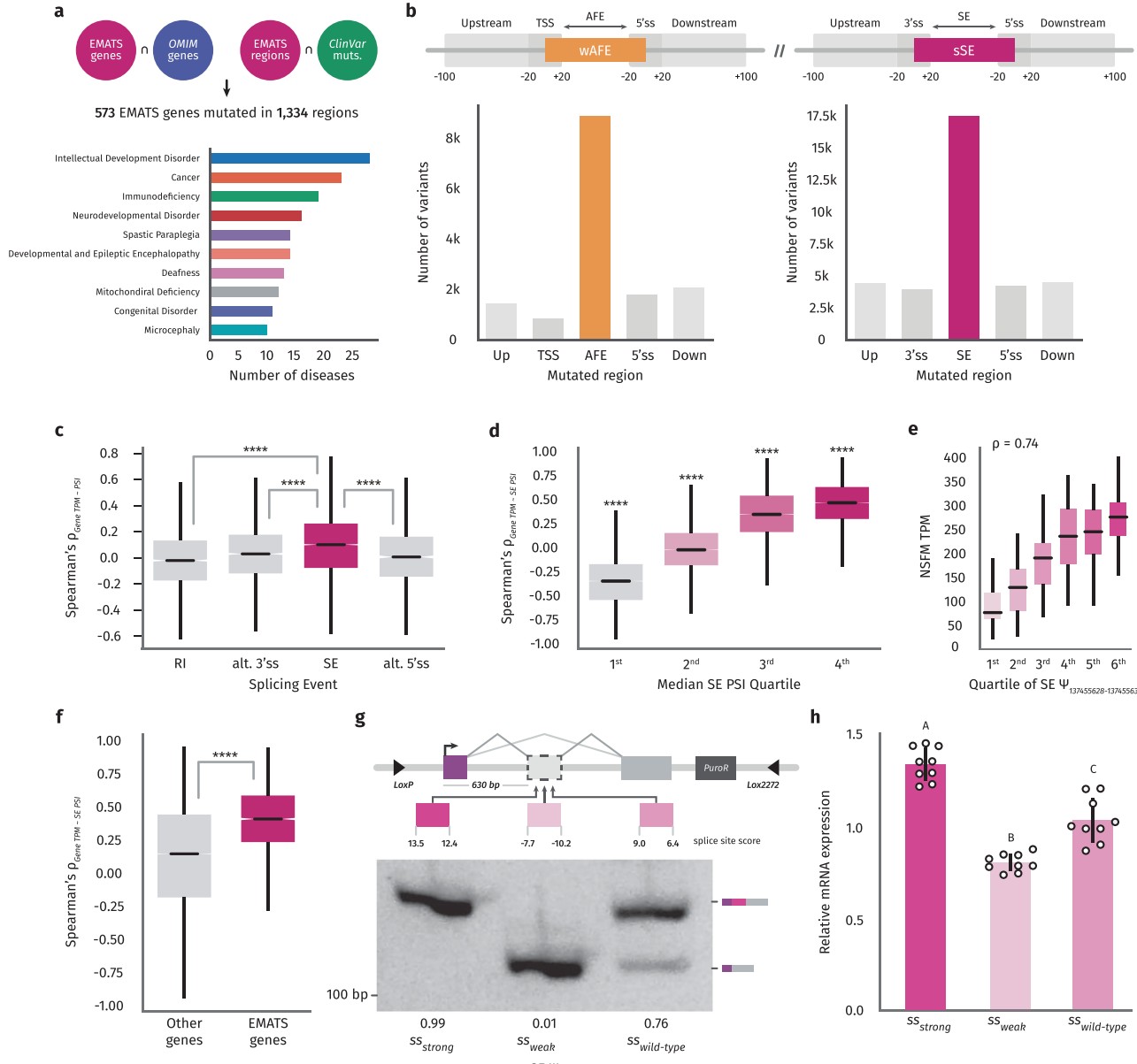

**Fig. 2 | Pathological variants and coupling between splicing and expression in EMATS genes. a** A step-by-step identification of Mendelian pathological genomic variants associated with EMATS regions of human EMATS genes (upper) and genetic human diseases associated with EMATS genes (lower). **b** Number of pathological genomic variants in human EMATS genes located in specific regions of EMATS exons. **c** Distribution of the spearman correlation between gene expression in TPMs and different splicing events: Intron retention ($n$ = 6852 events), alternative 3' splice sites ($n$ = 13,222 events), skipped exons ($n$ = 33,623 events), and alternative 5' splice sites ($n$ = 9,061 events) in PSI. **d** Distribution of spearman correlation between gene expression and SE PSI across human tissues and samples binned in quartiles of SE PSI; $n$ = 33,623 skipped exons. **e** Distribution of gene expression values in TPM for NSFM gene binned by sextiles of inclusion in PSI values for the SE in an EMATS structure across brain amygdala samples; $n$ = 152 biologically independent samples. **f** Distribution of the spearman correlation between gene

expression and SE PSI across human tissues and samples for non-EMATS genes compared to EMATS genes; n (other genes) = 55,864, and n (EMATS) = 2306. Statistical significance is indicated by asterisks (****$p$ < 0.0001; two-sided independent t-test). **g** Diagram of splicing reporter indicating the splice site scores of the alternative exon in the wild type construct and two mutants with different PSI values by RT-PCR. Quantification of densitometric analyses was carried out with Fiji, +Exon/-Exon ratios are shown at the bottom of each lane. **h** Expression of three splicing reporters—wild type, strong and weak splice sites (ss)—integrated in cells was evaluated by RT-qPCR after. The scatter plot with bars represents the mean ± standard error, and individual dots represent individual data points; $n$ = 3 experiments, with 3 biological replicates and 3 technical replicates. Bars with the same letter indicate no significant difference between means as determined by one-way ANOVA (Tukey's HSD test, $p$ < 0.0001) comparing across groups.

Fig. 2a, b). Notably, the association between splicing of skipped exons and gene expression levels is stronger in tissue-specific EMATS genes (Fig. 2f). To extend this analysis to other cellular transitions, we analyzed the associated changes in splicing ratios and AFE usage during SARS-CoV-2 infection. We observed that EMATS genes show a significantly stronger association between splicing of skipped exons and AFE usage during viral infection compared to non-EMATS genes

(Supplementary Fig. 2c), with the highest correlation occurring when there is an AFE located proximal to and upstream of the SE (Supplementary Fig. 2d). These observations indicate that splicing of SEs is associated with both promoter usage and gene expression in human transcriptomes, and their regulation is positively correlated during cellular transitions, with the strongest effects in EMATS genes. In order to test the potential impact of genetic mutations affecting splicing on

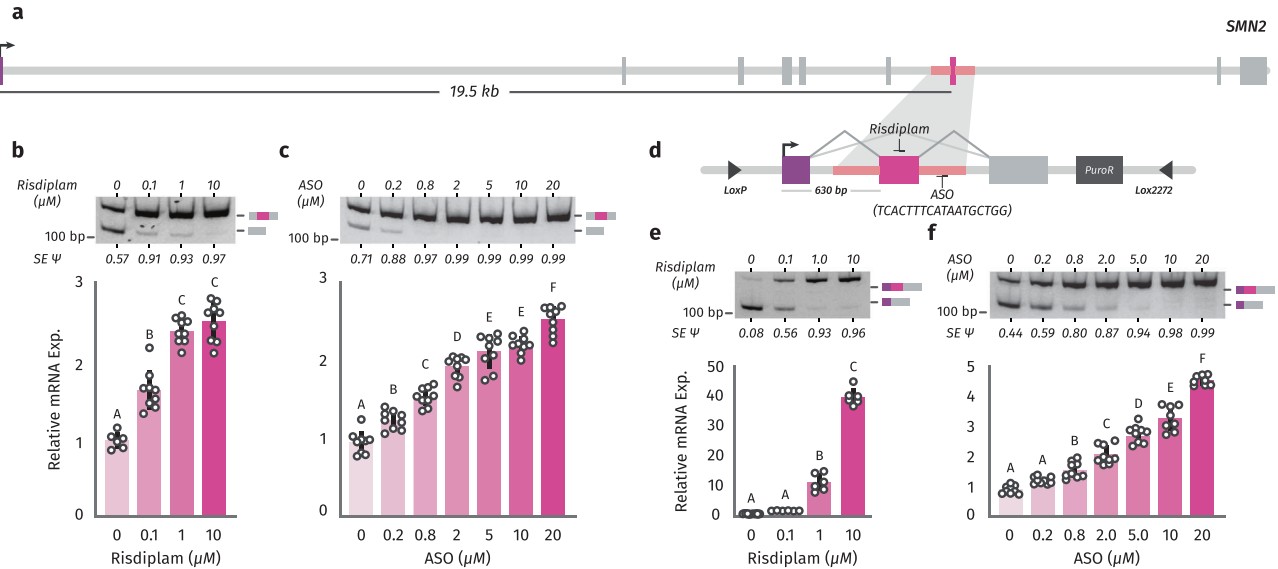

**Fig. 3 | Small molecules and ASOs that increase exon inclusion can boost gene expression. a, d** Diagram of the SMN2 gene (**a**) and SMN2 splicing reporter in the context of an integration plasmid used to generate HEK293T-A2 stable cells (**d**) highlighting the alternative exon 7 and the binding sites of Risdiplam and the ASO modeled after Spinraza. (**b, c, e, f**) Wild-type HEK293 cells (**b, c**) or stable HEK293T-A2 cells expressing the SMN2 splicing reporter (**e, f**) were treated with ethanol (-) or 0.1 μM, 1.0 μM, and 10 μM Risdiplam for 24 h (**b, e**) or a scramble ASO (-) or 0.2 μM, 0.8 μM, 2.0 μM, 5.0 μM, 10 μM, and 20 μM of the specific ASO for 24 h (**c, f**). Inclusion of alternative exon 7 in SMN2 was evaluated by RT-PCR (top).

Quantification of densitometric analyses was carried out with Fiji, and +Exon/-Exon ratios are shown at the bottom of each lane. SMN2 expression was evaluated by RT-qPCR (bottom). Results from at least three independent experiments are shown. The scatter plot with bars represents the mean ± standard error, and individual dots represent individual data points; $n = 3$ experiments, with up to 3 biological replicates and 3 technical replicates. Bars with the same letter indicate no significant difference between means, as determined by one-way ANOVA (Tukey's HSD test, $p < 0.0001$) comparing across treatments.

gene expression, we generated stable cell lines expressing a biochromatic splicing reporter, as well as different versions of the reporter in which the splice sites of the internal alternatively spliced exon were mutated to favor exon skipping (weak splice sites with negative MaxEnt scores) or exon inclusion (strong splice sites with maximum MaxEnt scores). We observed that exon skipping is associated with decreased mRNA levels of the host gene, while exon inclusion is associated with increased gene expression (Fig. 2g, h). These observations open up the possibility of therapeutic strategies by manipulating the expression of genes associated with human genetic diseases through changes in splicing.

## Splicing boosts gene expression

Since splicing of internal exons is associated with gene expression in humans, we wondered whether splicing perturbations can be used to control gene expression. Moreover, since the association between splicing and gene expression is stronger in EMATS genes, we explored the possibility of using EMATS to develop a therapeutic strategy to treat genetic diseases based on splicing modulation. We already showed that inhibition of splicing with ASOs can downregulate expression from the host genes through EMATS[13]. While several strategies have been developed to downregulate gene expression, successful strategies to activate gene expression remain to be developed for treatment of the thousands of diseases caused by null or reduced expression of specific genes. Here, we investigate whether splicing activation can be used to boost gene expression through EMATS as a potential therapeutic strategy. First, we used the endogenous SMN2 locus associated with muscular dystrophy (Fig. 3a) and triggered splicing changes with small molecules to evaluate potential modulations in gene expression levels[23]. Treatment with Risdiplam[20,21], a small molecule designed to upregulate splicing of SMN2 alternative exon 7, increased inclusion of the alternative exon up to 97% and triggered a significant increase in expression of the SMN2 gene (Fig. 3b). We next sought to explore whether ASOs targeting splicing have similar effects

as small molecules. We used an ASO modeled after Spinraza[22] that is known to upregulate splicing of SMN2 exon 7 by blocking an intronic splicing silencer located downstream of exon 7[32]. The Spinraza-like ASO was able to upregulate splicing of the endogenous SMN2 gene with all concentrations tested and increase gene expression levels to similar levels compared with the small molecule (Fig. 3c). This observation is consistent with the original study of SMN2 exon 7-targeting ASOs in which ASOs that promoted exon 7 inclusion of the endogenous locus increased full-length SMN protein levels[33] and demonstrates that small molecules and ASOs that increase inclusion of alternative exons can activate gene expression. However, the endogenous SMN2 gene does not support a coding transcript with EMATS structure, as the primary open reading frame requires initiation at a TSS > 10 kb upstream from the gene's skipped exon. To analyze the effect of splicing in upregulating gene expression in an EMATS-like locus, we generated stable cell lines expressing a splicing reporter based on SMN2 where exon 7 is located less than 1 kb downstream of a CMV promoter (Fig. 3d). Treatment with Risdiplam increased inclusion of the alternative exon in our system by several folds and triggered a ~45-fold increase in expression of the SMN2 reporter (Fig. 3e, Supplementary Fig. 3a). The Spinraza-like ASO was also able to upregulate splicing of SMN2 under the regulation of the cytomegalovirus (CMV) promoter for all concentrations tested and increase gene expression levels up to 5-fold with the highest concentration (Fig. 3f).

Both the ASO and small molecule drug triggered a gradual increase in gene expression with higher concentrations following a linear pattern with bigger effects on splicing associated with bigger effects on gene expression (Supplementary Fig. 3b). Overall, our findings demonstrate that small molecules and ASOs targeting splicing in EMATS-like genes are an effective method to activate gene expression.

## Splicing can activate expression from different promoters

In order to investigate whether the upregulation of gene expression by small molecules and ASOs that target splicing could be

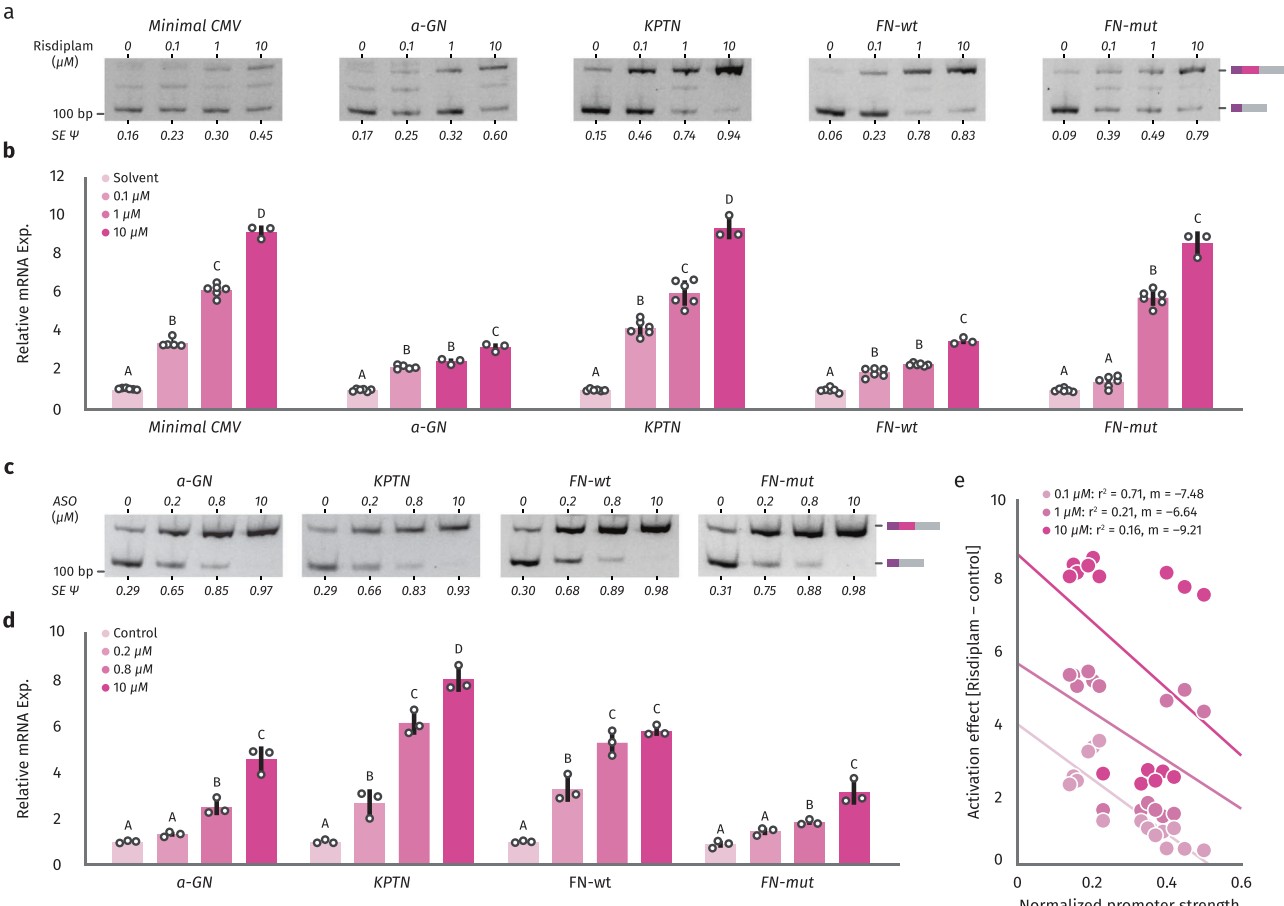

**Fig. 4 | Larger effects of splicing on gene expression with efficient exon inclusion and weak promoters. a–d** HEK293T-A2 stable cell lines expressing SMN2 splicing reporter under the regulation of different promoters (CMV, Minimal CMV, alpha-globin (a-GN), KTPN, Fibronectin wild type (FN-Wt), and Fibronectin mutated (FN-Mut)) were treated with Ethanol (-) or different concentrations of Risdiplam (0.1 μM, 1.0 μM, and 10 μM) for 24 h (**a, b**) or a scramble antisense oligonucleotide (-) or different concentrations of a specific ASO (0.2 μM, 0.8 μM, and 10 μM) for 24 h (**c, d**). The inclusion of alternative exon 7 in SMN2 was evaluated by RT-PCR and quantification of densitometry analyses was carried out with Fiji, and the +Exon/-Exon ratios are shown at the bottom of each lane (**a, c**). SMN2 expression was evaluated by RT-qPCR (**b, d**). Results from three independent experiments are shown. The scatter plot with bars represents the mean ± standard error, and individual dots represent individual data points. Bars with the same letter indicate no significant difference between means as determined by Two-way ANOVA with corrections for multiple comparisons ($p < 0.0001$) comparing each treatment mean with each control per promoter. **e** Linear regression of gene expression changes in SMN2 splicing reporter following Risdiplam treatment with different doses for stable lines under the regulation of different human promoters and promoter strength measured by basal RT-qPCR in control conditions.

generalized to other promoter sequences, we performed similar experiments using the splicing reporter under the regulation of different promoter sequences. For these experiments, we used three natural human promoter sequences of different strength (alpha-globin, Fibronectin, and KPTN) and include two mutant versions (a minimal CMV promoter containing only 39 bp, and a mutant of the Fibronectin natural promoter in which the CRE at position −170 and the CCAAT box at position −150 have been disrupted by introducing point mutations that abolish binding of the corresponding transcription factors[34]). We observed an increase in inclusion of the SMN2 alternative exon with both Risdiplam and ASO treatment under the regulation of all natural promoter sequences and mutants tested (Fig. 4a, c). Like the effect under the CMV promoter, the splicing regulation increased with higher drug/ASO concentrations for all promoters tested. In all cases, the upregulation of SMN2 exon 7 inclusion was associated with an increase in gene expression levels of our reporter (Fig. 4b, d).

As discussed previously, the EMATS effect is stronger when the splicing of the alternative exon is highly efficient and the proximal promoter is weak. To analyze whether the splicing-mediated effect of small molecules/ASOs on gene expression follows the same rules, we quantified the drug-induced effect on gene expression for all different promoters tested and all different drug/ASO concentrations. We found that the effect of splicing on gene expression is stronger for higher drug/ASO concentrations which are associated with higher splicing efficiency (Supplementary Fig. 4a–e), suggesting that efficient splicing is associated with a larger effect on gene expression. Moreover, we found that the effect of small molecules and ASOs on gene expression followed a linear pattern with larger effects associated with weaker human promoters (Fig. 4e, Supplementary Fig 4f). Thus, consistent with our previous analyses, the effects of splicing on transcription are stronger when splicing is efficient and promoters are weak. Together, these findings indicate that upregulation of splicing of internal exons with small molecules and ASOs can be used to activate gene expression levels independently of the promoter used, but the strongest effects are induced with weaker human promoters.

## EMATS acts at the transcriptional level

In order to identify the step of gene expression that is modulated by splicing changes, we analyzed newly synthesized RNA levels of our splicing reporter by metabolic labelling with 4-thiouridine (4sU)

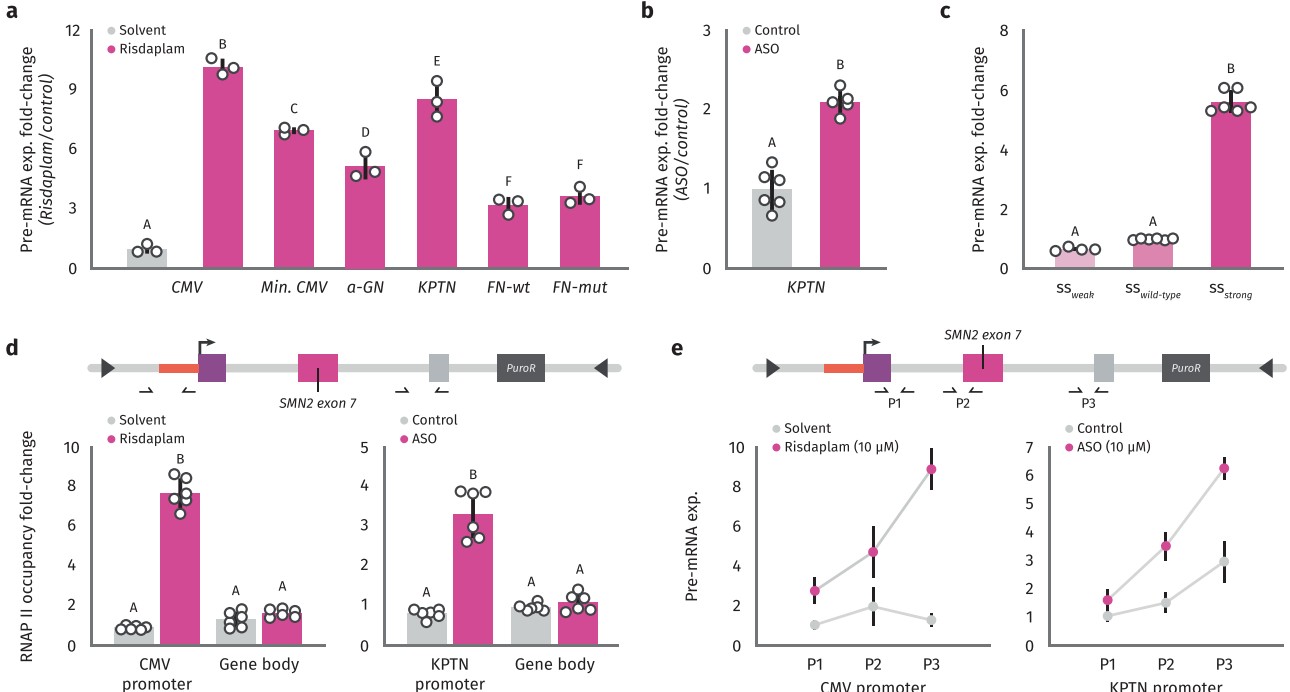

**Fig. 5 | EMATS activates transcription initiation and elongation. a**, **b**, **c** Newly synthesized RNA was labelled with a 30 min pulse of 4sU and extracted from HEK293T-A2 stable cell lines under the regulation of different promoters after 30 min treatment with Risdiplam (10 uM) (**a**), the ASO (10 uM) (**b**), or splicing reporters with different splice sites (wild type, strong and weak splice sites (ss)) (**c**). SMN2 expression was evaluated by RT-qPCR. **d** RNA polymerase occupancy at the promoter region or gene body was evaluated by CUT&RUN-qPCR using a specific antibody (Rpb1 CTD) in HEK293T-A2 stable cell lines expressing the SMN2 splicing reporter and treated with Risdiplam 10 uM (left) or the ASO 10 uM (right).

**e** Transcription elongation rate were approximately measured by calculating expression at proximal and distal regions of the SMN2 splicing reporter in HEK293T-A2 stable cell lines following a 30 min metabolically labelling pulse with 4sU. The scatter plot with bars represents the mean ± standard error, and individual dots represent individual data points; $n = 3$ experiments, with up to 3 biological replicates and 3 technical replicates. Bars with the same letter indicate no significant difference between means as determined by Two-way ANOVA with corrections for multiple comparisons, comparing each treatment mean with each control per promoter ($p < 0.0001$).

following splicing activation with the highest drug concentration, ASO doses, and splice site mutations. We observed a significant effect of splicing on newly synthesized RNA levels of the reporter with all splicing activation methods (Fig. 5a–c). Notably, the magnitude of the effect varied with promoter sequences. For some promoters, the effect on nascent RNA was weaker compared to the effect on steady-state mRNA levels, while it was stronger for other promoters (Fig. 5a). These findings suggest that while the effect of splicing on gene expression induced by small molecule drugs is at the transcriptional level, downstream steps of gene regulation may also play a role, and the magnitude of that role depends on the promoter sequence. As noted for steady-state RNA, small molecules that activate splicing also boost transcription initiation with the highest effects observed for weak promoters (Supplementary Fig. 5a).

To explore further the transcriptional steps regulated by splicing, we quantified RNA polymerase II occupancy levels at the integrated reporter with CUT&RUN-qPCR experiments following treatment with the small molecule drug and ASO. We observed a significant increase in RNA polymerase II occupancy levels at the promoter regions, with both methods favoring exon inclusion but resulting in no changes in the gene body (Fig. 5d). These observations suggest a splicing-dependent activation of transcription initiation but do not rule out activation of other transcriptional levels. Indeed, we investigated whether transcriptional elongation levels were affected by splicing in our system and observed a significant increase in RNA polymerase II elongation rate following splicing activation with both the small molecule and the ASO (Fig. 5e). Together, these findings indicate that EMATS works at the transcriptional level by boosting transcription initiation and elongation, although downstream steps of gene regulation may also contribute to the effect.

## Splicing boosts transcription from nearby promoters through interactions with transcription machinery

Our original characterization of EMATS suggested that not only the strength of the promoter and inclusion levels of alternative exons contribute to the effect but that the relationship is also a function of the proximity between the promoter and internal exon. To test this, we cloned natural intronic sequences of different lengths into our integrated splicing reporter. Although we introduced new intronic sequences, we did not observe any new or cryptic splice sites and therefore did not detect any changes in splicing products (Supplementary Fig. 6a). Interestingly, we found that the ability of both the small molecule drug and ASO to activate expression depended on the distance between the promoter and alternative exon. Specifically, we observed the strongest effect for the shortest distance (~500 nt), whereas the effect decreased by 2- to 3-fold for longer distances (~2.5 and 5.5 kb) (Fig. 6a, Supplementary Fig. 6b, c). This finding suggests that the distance between the promoter and alternative exon plays a critical role in determining the effectiveness of splicing-dependent expression modulators.

Finally, to explore possible factors involved in this phenomenon, we analyzed the correlation between gene expression and inclusion of skipped exons using RNA-seq data from a recent ENCODE project, which involved the knock-down of over 250 RNA binding proteins (RBPs). Our analysis revealed a positive association between gene expression and inclusion of skipped exons, consistent with our previous findings across human tissues. Importantly, this positive association was lost following knock-down of several RBPs (Fig. 6b).

To identify the RBPs that contribute most strongly to this association, we selected those with the largest residuals from the linear regression between global splicing and gene expression changes

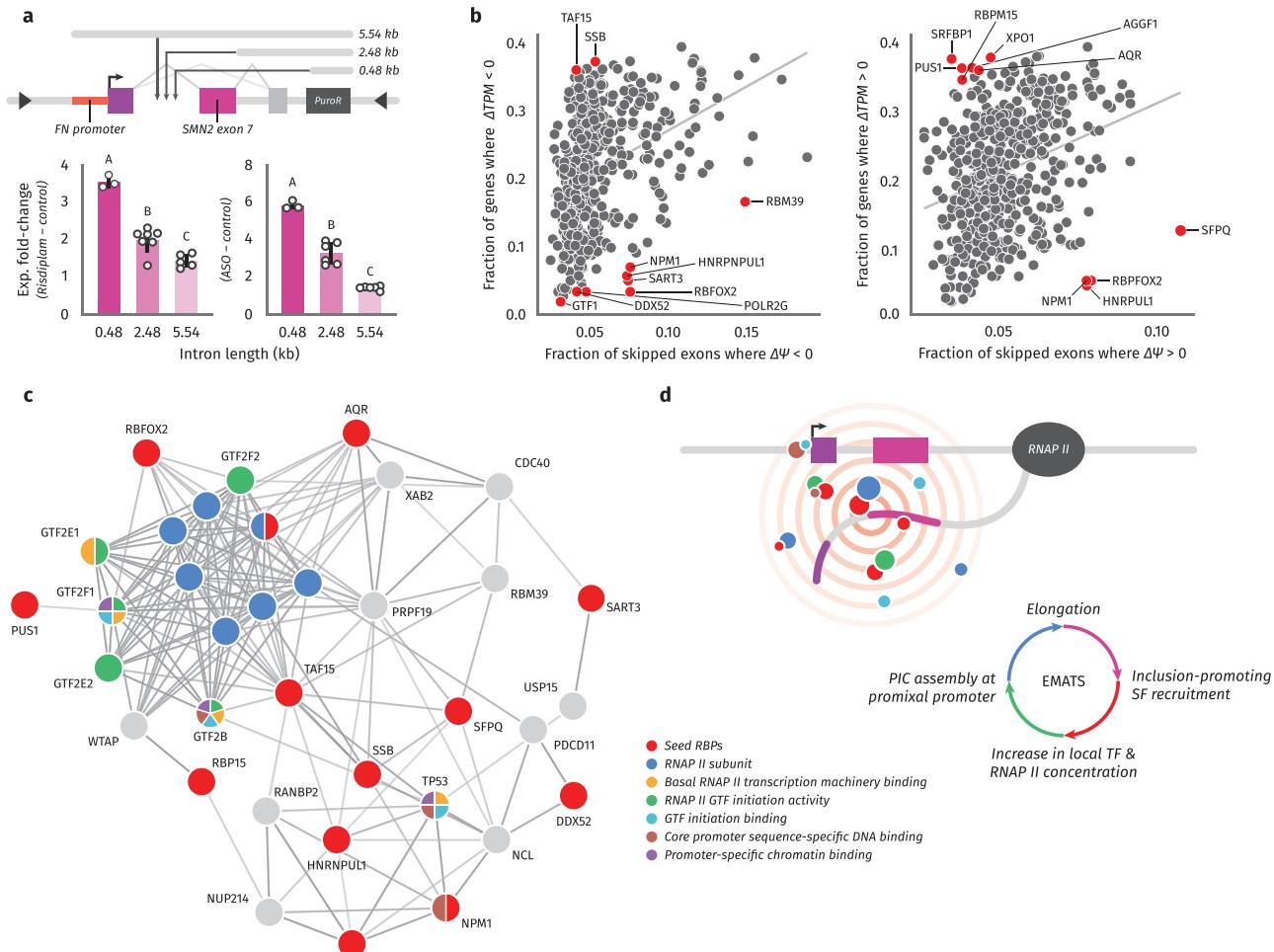

**Fig. 6 | EMATS acts at proximate genomic distances. a** HEK293T-A2 stable cell lines expressing SMN2 splicing reporter under the regulation of Fibronectin (FN) promoter and three different intronic sequences were treated with Risdiplam (left) or a scramble antisense oligonucleotide (-) or a specific ASO (+) for 24 h (right). Expression of the SMN2 reporter was evaluated by RT-qPCR. The scatter plot with bars represents the mean ± standard error, and individual dots represent individual data points; $n = 3$ experiments, with up to 3 biological replicates and 3 technical replicates. Bars with the same letter indicate no significant difference between means, as determined by one-way ANOVA (Tukey's HSD test, $p < 0.0001$) comparing across groups. **b** Correlation between the fraction of differentially upregulated (left) and downregulated (right) genes and the fraction of differentially upregulated or downregulated skipped exons following RBP knock-down. RBPs with largest residuals are highlighted in red. **c** PPI network for the 17 unique RBPs from (**b**) with the largest residuals, colored by Gene Ontology category (5 RBPs we not connected to the network and excluded from the visualization). PPI data are from the STRING database. **d** Model showing that splicing can activate transcription initiation depending on three main factors: the strength of the promoter, the inclusion of the skipped exon, and the genomic distance between them. Splicing factors (SF) recruited to the splicing event interact with transcription factors (TF) increasing the local concentration of TFs and inducing transcription from proximal promoters which, in turn, favors inclusion of skipped exons.

(Fig. 6b, Supplementary Fig. 6d). We then built a protein-protein interaction network for these RBPs and found that they primarily interact with transcription factors (Fig. 6c). These results suggest a possible local interaction model, in which RBPs recruited during splicing interact with transcription factors to jointly impact the expression and splicing of the same genes (Fig. 6d). Altogether, our findings demonstrate that small molecules and ASOs that activate splicing of EMATS-like isoforms can be used to boost gene expression at the transcriptional level from different human promoters, showing the largest effects for higher splicing efficiencies, weaker promoter sequences, and proximity between promoters and alternatively spliced exons.

## Discussion

Here, we provide a comprehensive list of human EMATS genes and establish their link to Mendelian diseases. EMATS genes are defined as protein-coding genes with highly included skipped exons located less than 5 kb downstream of a weak alternative promoter[13]. Using this definition, we identified thousands of genetic human variants that disrupt splicing of SEs in EMATS genes. Since the splicing of an SE can control the usage of alternative promoters in EMATS genes, we predicted that the effect of those genetic variants on splicing is magnified by a secondary effect of splicing-mediated perturbation of promoter usage and gene expression levels.

Connections between splicing and gene expression regulation across tissues and species have been observed. Our findings support further broadening of this connection, suggesting a direct control of gene expression through splicing of internal exons specifically positioned in EMATS-like genes. In a previous study, we had shown that the mechanism behind the exon-mediated gene expression control might be associated with direct recruitment of transcription machinery to nearby upstream promoters through splicing factors[13]. Here, we reinforced this hypothesis with a splicing reporter integrated in human cells, showing that the modulation of splicing with small molecules and ASOs has a direct effect on transcription.

We previously have shown that splicing inhibition of a skipped exon in an EMATS gene by splice site-targeting mutations or ASOs can

downregulate gene expression of the host gene by several folds. Over the past decade, several techniques have been developed to downregulate the expression of disease relevant genes. The most well-characterized technique is the post-transcriptional gene silencing mechanism known as RNA interference (RNAi) triggered by double-stranded RNA (dsRNA), which induces the formation of a ribonucleoprotein complex that mediates sequence-specific cleavage of the cognate transcript with the input dsRNA[35]. Less common downregulation techniques include the usage of artificial microRNAs, recombinant nucleic acid molecules with hairpin structures, cationic polymers and single-stranded ribonucleotide oligomers[36], and U1snRNP adaptors[37]. However, only a few strategies have been proposed to upregulate gene expression with therapeutic benefits. Since in most of the diseases associated with EMATS genes studied here, and several non-Mendelian diseases, gene upregulation would provide a more powerful strategy than gene downregulation, we tested our system as a therapeutic alternative to increase gene expression through splicing activation. Using a small molecule that increases inclusion of SMN2 exon 7, we were able to upregulate expression of the host gene up to 45-fold. Our strategy is complementary to the one developed by Lim et al. 2020[19], as both aim to target splicing to activate expression of a different subsets of genes. While Lim et al., 2020's method works for NMD-inducing alternative splicing events, our method produces the strongest effects in EMATS genes. Newly synthesized RNA analyses and RNA polymerase II occupancy measurements showed that this effect is partially explained by an increase in transcription initiation and elongation, but does not rule out contributions from downstream steps of gene regulation. Since previous studies indicate that the effect of splicing on transcription is magnified by a secondary effect on translation efficiency[13], we expect an even stronger effect at the protein level.

Here, we demonstrate that the splicing-mediated activation of gene expression through EMATS depends on three main factors: the strength of the promoter, the inclusion of the skipped exon, and the genomic distance between them. Altogether, our findings provide evidence for the development of a therapeutic strategy to increase gene expression through splicing by EMATS, a comprehensive list of genes that are sensitive to this approach, and the conditions in which this approach produces the strongest effects.

## Methods

### Plasmids
To generate plasmids with different splice sites, we used the plasmid pEM68941[38] containing the wild-type (WT) splice site as a template. We designed strong and weak splice site sequences using MaxEntScan[39] (Supplemental information, Supplementary Table 1) and incorporated these sequences into the plasmid pEM68941 using Gibson assembly methodology (NEB, E2611L). Linearized pEM68941 was prepared using the restriction enzymes *Cla*I:*Xho*I (ThermoFisher, ER0141 and ER0692). To build the SMN2 plasmids, a 400 bp region containing alternative exon 7 (54 bp) of SMN2 gene flanked by intronic sequences of 173 bp at each end was cloned under the regulation of different promoter sequences: Cytomegalovirus (CMV), CMV minimal (Mini- CMV), alpha globin (a-GN), KPTN, fibronectin (FN-Wt), and mutated fibronectin[34] (Supplemental Information, Supplementary Table 1) into the plasmid pEM689[38] using Gibson assembly methodology (NEB, E2611L). pEM689 was linearized with restriction enzymes *Sphe*I:*Nhe*I (ThermoFisher, ER1251 and ER0971). To construct the plasmids with introns of different sizes, we used the plasmids containing the different promoters as described above. The sequence for the 2.04 kb intron was taken from the genomic region (GRCh38/hg38) chr5:70,053,253-70,055,563, and the sequence for the 5.1 kb intron was taken from the genomic region chr5:70,930,233-70,935,333; these sequences were amplified using specific primers (Supplemental information, Supplementary Table 1). The PCR products were cloned into the plasmid pEM8941, the

different promoter plasmids, using Gibson assembly methodology. Prior to cloning, pEM8941 was linearized using the restriction enzyme *Cla*I (ThermoFisher, ER0141).

### Cell line, culture, stable transfection and electroporation
HEK293T-A2[38] cells were grown in DMEM with high glucose and pyruvate (Gibco, 11965118), supplemented with 10% fetal bovine serum (FBS, Gibco, A31406-02). To generate stable cells with genomic integration of the splicing reporter, the different plasmids were individually co-transfected with 10% (wt/wt) of a Cre-encoding plasmid; a total of 1.5 mg plasmid was transfected into each well of a 6-well culture plate using Lipofectamine 3000 (ThermoFisher, L3000-015) and Opti-MEM (Gibco, 31985-070) according to the manufacturer's recommendations. After transfection with the splicing reporter, cells were selected with puromycin (Gibco, A1113802) at 1 ug/mL for 2 weeks. Cells were treated with Risdiplam (MedChemExpress, HY-109101, soluble in ethanol) at various concentrations (0.1 μM, 1 μM, and 10 μM) for 24 h. Control cells were treated with ethanol. For antisense oligonucleotide (ASO) treatment (Gene Tools), 1 million cells per condition were electroporated with 0.2, 0.8, 2.0, 5.0, 10, and 20 μM ASO targeting SMN2 (TCACTTTCATAATGCTGG) using the MaxCyte ATX and the Optimization 4 program, following the manufacturer's instructions. Stable cell lines were plated in 6-well plates and incubated for 24 h.

### RNA extraction, nascent RNA, RT-PCR and qPCR
Total RNA was extracted using TRizol (Invitrogen) and equal quantity from each sample were used, according to the manufacturer's protocol. Reverse transcription was performed in a reaction mix containing 1 μg of total RNA, using cDNA Synthesis Kit (Thermo Scientifc) according to the manufacturer's instructions. For nascent RNA extraction, RNA was metabolically labeled with 4-thiouridine (4sU, Sigma-Aldrich) for 30 min and labeled RNA was extracted and amplified according the protocol described by Bernd Rädle[40]. Quantitative PCR analyses were performed with SYBR green labeling (Thermo Scientific Maxima SYBR Green/ROX qPCR Master Mix (2X), K0222) using a ABI7900HT Fast Real-Time PCR System (Applied Biosystems). Relative expression level of each analyzed gene was calculated by $2^{-\Delta Ct}$, where $\Delta Ct$= (Ct target gene - Ct control gene), using *GAPDH* as an internal control[41] (Supplemental Information, Supplementary Table 1). Transcript abundance quantification was measured by technical replicates, three biological replicates, and at least three independent experiments. CUT&RUN-qPCR experiments followed the nuclei isolation protocol described by Panday, 2022[42]. One million cells per condition were used. Concanavalin A beads (Bangs Laboratories, Cat No. B P531) were activated (10 μL of beads per sample) and washed 2X with binding buffer and resuspended in a binding buffer. 10 μL of prepared ConA bead was added to each sample, and cells were resuspended. Samples were placed in a mixer for 15 min at room temperature. For cell permeabilization and primary antibody binding, the tubes were placed on a magnet stand and the liquid was removed. Then, 400 μL of chilled freshly prepared antibody buffer was added. Rpb1 CTD antibody (Cell signaling, Cat No. 2629) was added to each tube (1:50–8 μl), mixed, and incubated on a Nutator rocking platform at 4 °C overnight. For pA/G-MNase binding, cells were washed twice with chilled digitonin buffer, resuspended in 350 μL of digitonin buffer, and 2.5 μL of pAG-MNase (Epicypher, Cat. No. 15–1016) was added. Each sample was then placed on a Nutator for 30 min at room temperature. For targeted chromatin digestion and release, cells were washed twice with chilled digitonin buffer, and 300 μL of chilled digitonin buffer and 3 μL of chilled 100 mM CaCl2 were added to activate MNase. Samples were rocked on a Nutator at 4 °C for 4 h, and after that, 100 μL of chilled stop buffer was added to each tube. DNA was extracted from the supernatant using a Qiagen MinElute PCR purification Kit, following the manufacturer's instructions. Finally, qPCR was performed using

specific primers (Supplementary Table 1 in the Supplementary Information). For the elongation rate measurement, we adapted previous methods[43,44] and cells were treated with 10 μM Risdiplam or 10 μM ASO for 24 h. RNA was metabolically labeled with 4-thiouridine (4sU, Sigma-Aldrich) for 30 min, and labeled RNA was extracted as mentioned above. Reverse transcriptase reaction was initiated with random decamers. Quantification of the pre-mRNAs was performed by real-time PCR with amplicons spanning the intron-exon junctions, using the primers listed in Supplementary Table 1.

## Quantification and statistical analyses

**Identifying and characterizing of EMATS genes.** To identify EMATS genes, we collected protein-coding genes from the chromosomal hg38 annotation set (GENCODE release 26) and reduced the gene set using the following criteria:

1. the gene must contain a highly included skipped exon (SE), where highly included is a median percent-spliced in (PSI) value greater than the dataset-wide median of all SE PSI values;
2. the gene must contain a weak alternative first exon (AFE), where weak is a median PSI value less than the dataset-wide median of all AFE PSI values; and
3. the strong SE and weak AFE must not overlap and have 5' coordinates within 5 kilobases.

The AFE PSI values were calculated with the HIT index pipeline[26], adapted to exclude unannotated first exons. Briefly, the pipeline inspects an exon's splice junction read (SJR) profile for a downstream SJR imbalance, modelling other characteristic SJR profiles to then allow comparison and confident first exon calling. The default and recommended parameters were used. rMATS[27] was used to calculate SE PSI values and outputs derived from SJR counts were used for analysis; unannotated skipped exons were excluded from downstream analyses. The default parameters (apart from those indicating paired-end data and read length) were used. Detailed code to identify EMATS genes is provided at https://github.com/fiszbein-lab/emats-genes.

**EMATS gene architecture.** The analyses in Fig. 1 are described below:

- Given a 0-based, half-open coordinate system, a gene's kilobase length is defined as $\frac{ending\ coordinate - starting\ coordinate}{1000}$.
- Given a 0-based, half-open coordinate system, an exon's distance to its host gene's 5' coordinate is defined as $exon_{start} - gene_{start}$ for forward-strand features and $gene_{end} - exon_{end}$ for reverse-strand features.
- To explore EMATS at the tissue level, we applied the above criteria to each set of samples comprising a particular tissue. Tissue-specific EMATS genes were called when a gene was identified in the main, focal tissue and no others, e.g., a brain-specific EMATS gene may be identified in multiple brain tissue sub-types but no other tissues.

**EMATS genes involved in human diseases.** To investigate diseases associated with EMATS genes, the *Online Mendelian Inheritance in Man* morbid map (generated August 4th, 2022; https://omim.org) and *ClinVar* database (generated July 22nd, 2022[45]) were merged and then intersected with the weak AFE and strong SE regions in EMATS genes.

**Gene ontology.** Functional, molecular classification of EMATS genes was performed with the PANTHER web application (version 17.0)[46].

**Alternative splicing and gene expression.** To investigate correlations between alternative splicing and gene expression, we identified alternative splicing events with rMATS[27], used gene abundances publicly-available from the GTEx project, and merged the outputs, dropping samples when the splicing event exhibited constitutive inclusion or exclusion. If at least 70 paired observations were available, Spearman

correlations between an event's percent spliced-in values and its host gene's transcripts-per-million values were then calculated with SciPy's stats.spearmanr function; the resulting splicing event, locus-specific rho values were then merged at the gene level and, when applicable, again at the level of the splicing event. Linear regression was performed with SciPy's stats.linregress function.

**Splicing changes during SARS-CoV-2 infection.** To examine RNA splicing changes during SARS-CoV-2 infection, we obtained RNA-sequence data from 3 mock and 3 SARS-CoV-2 infected NHBE (Normal human Bronchial Epithelial) cell samples[47]. The sequence data were aligned to the GRCh38 genome using STAR[48]. Levels of gene expression was analyzed using Kallisto[49] with an FDR of 0.05 and DESeq2[50]. Internal splicing changes (SE, alternative 5' splice site (A5SS), alterative 3' splice site (A3SS), mutually exclusive exon (MXE), and retained intron (RI)) were identified using rMATS[27] using exon and junction counts. The HIT index pipeline was used to analyze AFEs PSIs and DEXSeq[51,52] was used to account for dispersion properties and an FDR < 0.05.

## RBP and protein-protein interaction analyses

The RBP analyses utilize the 255 RBP knock-down dataset generated by the Gravely lab for the ENCODE project[53,54]. Each knock-down experiment has two replicates, as well as an associated, two-replicate control experiment. Using expected RSEM[55] gene quantifications provided by the ENCODE project, as well each experiment's pre-aligned RNA-seq samples, we identified differential expression and splicing with edgeR[56] and rMATS[27], respectively, using the tools' default parameters. A significant (i.e., a multiple-testing corrected *p*-value less than 0.05), differential gene expression or exon skipping event was considered upregulated when the difference between knock-down and control was greater than 0 and downregulated when the difference was less than 0. The fraction of genes that met either criteria in each experiment were then regressed on the fraction of skipped exons that met the same criteria, where the fractions were computed against the total number of genes or exons reported in the respective tools' output. The 17 RBPs with the largest residuals from this regression were used to construct a protein-protein interaction network with medium confidence set as the minimum interaction score; a maximum of 20 1st-shell interactors; and 0 2nd-shell interactors. Molecular gene ontology functions relating to transcription initiation were then highlighted in the web application's analyses tab.

## Software for data analysis and visualization

For data analysis, we used BEDTools (v2.30.0)[57], SamTools (v1.12)[58], GenomicRanges (v3.17)[59] and the UCSC Genome Browser[60]. Statistical analyses were performed in R (v.1.3.1056 and v.4.1.3), Stata (v.16.1), Prism 8 (v.3.4.2), and SciPy (v.18.1)[61]. Visualization was performed with the R package ggplot[62] and Python (v.3.9.12) libraries pandas (v.1.4.2)[63], matplotlib (v.3.5.2)[64], and seaborn (v.0.11.2)[65]. Lower and upper hinges of box plots correspond to the 25th and 75th percentiles, respectively. The upper and lower whiskers extend from the hinge to the largest and lowest value no further than 1.5 × IQR (interquartile range), respectively. Statistical significance is indicated by asterisks (*$p < 0.05$, **$p < 0.01$, ***$p < 0.001$, ****$p < 0.0001$, *****$p < 0.00001$).

## Reporting summary

Further information on research design is available in the Nature Portfolio Reporting Summary linked to this article.

# Data availability

The data that support this study are available from the corresponding author upon reasonable request. To identify EMATS genes, we used all 17,250 samples available on the AnVIL repository included in the GTEx version 8 release. Source data are provided with this paper.

## Code availability

The code used to identify EMATS genes is available at https://github.com/fiszbein-lab/emats-genes (https://doi.org/10.5281/zenodo.7942478).

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

## Acknowledgements

We thank Christopher B. Burge for helpful discussions, ideas, and comments, as well as members of the Fiszbein lab for feedback on the manuscript. This work was supported by university funds and grants to A.F. from the NIH (1R35GM147254-01), the Richard and Susan Smith Family Foundation, and the Pew Latin American Fellows Program in the Biomedical Sciences to M.U.A.

## Author contributions

M.U.A. performed all experiments (Figs. 2–6). S.T.M. developed the code to identify EMATS genes and performed all computational analyses (Figs. 1, 2, and 6). Z.S. developed a first version of the code to identify EMATS genes. R.R. performed a preliminary analysis for Fig. 2. A.F. designed the study, wrote the manuscript, and supervised the work.

## Competing interests

Ana Fiszbein and Christopher B. Burge have a patent related to their work on EMATS (U.S. patent 62/740,881 and International Patent Application PCT/US2019/044936, MIT patent #20890). The remaining authors declare no competing interests.
