## [Peer Review File · Nature Communications]

Splicing activates transcription from weak promoters
upstream of alternative exonsREVIEWER COMMENTS

Reviewer #1 (Remarks to the Author):

This manuscript by Uriostegui-Arcos and colleagues deals with the recently uncovered link that ties alternative splicing with transcriptional activation. Genes where this link exists and the architecture fits (highly included skipped exons located less than 5 kb downstream of a weak alternative promoter) are called EMATS genes. EMATS is an interesting concept that requires careful examination. The study provides a step in this direction and points out many associations between alternative splicing of EMATS genes and stimulation of their transcription. While there may be a subclass of alternatively spliced genes whose structure may be more conducive to transcriptional co-regulation, it is not clear whether architectural parameters will be the most revealing indicators as to how EMATS functions. Overall, I find the study interesting but lacking in clarity and requiring more work: more EMATS genes need to be investigated and more progress is required on molecular mechanisms of action.

1. What is the evidence that transcription is stimulated? In Figure 4 for example, could the increase in steady-state SMN2 mRNA be caused by increased RNA stability. Investigating RNA polymerase occupancy and elongation rate would be needed to add credence to the promoter-linked stimulation of gene expression.

2. Fig. 4b. Based on the RT-PCR which measures AS, the increase in exon inclusion in SMN2 was certainly not 100-fold as stated given that the PSI went from 0.20 to 0.97 with the highest concentration of risdiplam. If that is the case, I worry that the numbers given about increased SMN2 expression (45-fold and later claims) are also overstatements. While the RT-PCR gels shown are not quantitative and not designed to measure global expression, one would expect the images to display some increase in overall expression, which they are not.

3. Given that the significant association between skipped exons and gene expression is barely better for EMATS genes (page 8), there is a need to analyze genes that are not in the EMATS category to compare their response with small molecules, ASOs and different promoters. Several examples of EMATS genes would also be required to allow generalization.

4. How does the impact of risdiplam on SMN2 splicing and gene expression compare with that of the ASO? i.e., does a similar change in PSI yield a similar stimulation of mRNA expression?

5. The claim that genetic variants reprogram splicing and gene expression should be investigated more thoroughly using reporter minigenes.

6. An alternative and possibly more mundane explanation for EMATS may be that EMATS genes are in general longer producing longer transcripts. Such long transcripts are likely being bound by more factors and have more introns. Thus, they may have more time to stochastically move to areas of the nucleus where splicing and transcription factors exist at higher concentrations. Likewise, improving spliceosome interactions of alternatively spliced transcripts that are still in association with their DNA may increase the residency time of such transcripts in areas rich in transcription factors, hence explaining stimulation.

Other comments

1. Not quite sure what Supplementary Table 2 is, and no description is offered. May be indicate the names of genes along with their coordinates.

2. Intro: What is a "highly included skipped exon"?

3. EMATS must be rare enough to explain why transcription regulation and alternative splicing regulation have always been considered distinct layers of regulation (i.e., genes displaying tissue-

specific AS regulation are not the one displaying tissue-specific transcriptional regulation)? Does its actual prevalence match this prediction?

4. Fig. 2c, what are the two sets of numbers in FLH1B? When referring to patients with genetic variants in FLH1, the difference in the expression of each variant relative to normal should be clearly stated to clarify what exactly the mutations elicit. Similarly for MBOAT7, it is not clear from the text what the mutation promotes.

5. Only the best cases are shown in Fig. 3c. What is the range of R2 for all EMATS investigated?

6. Page numbering would be useful

7. Fig. S1a and S1b are reversed relative to the text.

8. Why isn't the 5'ss of SE not indicated as having a significant number of variations when the 3'ss is? (see Fig. 2b).

9. For FHL1, does the "downregulation of the second TSS by EMATS" the same as a downregulation of FHL1B? Please indicate what wAFE and sSE stand for in the figure legend?

10. Page 8: reference needed for association of new exons with increased expression.

11. SARS-Covid2 is not a term (Suppl Fig. 2)

Reviewer #2 (Remarks to the Author):

Splicing-dependent transcriptional activation

SUMMARY: In this study, Uriostegui-Arcos et al. systematically catalog genes that can undergo exon-mediated activation of transcription starts (EMATS) and investigate their potential to be transcriptionally activated by splicing modulation. They computationally identify two genes where pathogenic variants might disrupt protein domains via EMATS. Using a minigene reporter of SMN2 splicing, they show that both ASOs and small molecules that promote inclusion of exon 7 boost the expression of the reporter. These results are promising and suggest the possibility of EMATS as a mechanism underlying disease, as well as its potential for combating haploinsufficiency based genetic diseases. However, there are several limitations to this study that reduce my enthusiasm.

The authors make claims throughout the manuscript that overreach what their data demonstrates. For example, they claim that their Fig 2 "provide(s) evidence for protein structure disruption in specific EMATS genes caused by genetic variants". From my understanding, Fig 2 shows how specific genetic variants in these predicted EMATS genes could theoretically impact splicing and disrupt protein domains. However, the authors state that "Patients with these genetic variants express a wild type FHL1C, a mutated version of FHL1B which disrupts a highly conserved double zinc finger domain called LIM domain33, and reduced levels of a mutated version of FHL1A by EMATS". Isn't this a prediction based on the splicing score? Have the authors measured the observed splicing pattern for these genes in patients with mutations? If they do not have such data (and it is not clear from the description that they do), they should not make a strong claim as to the "expression" of various splice isoforms in patients. They also do not show any experimental data to conclusively connect this phenomenon to EMATS.

Another major weakness of this manuscript is that the authors do not address how much of the observed upregulation of gene expression is due to EMATS vs other phenomenon including enhanced stability or transport. While they acknowledge this limitation in one part of the manuscript, they routinely conflate increased mRNA level with enhanced transcription initiation due to EMATS throughout the paper. The only piece of data to suggest transcriptional upregulation is in the last figure where they measure nascent RNA level of the SMN2 reporter. The authors should more carefully account for the relative contribution of transcription induction through additional metabolic labeling experiments.

Finally, the authors need to demonstrate this phenomenon in an endogenous EMATS gene. All of their conclusions are based on an integrated minigene reporter and it limits the scope of the claims they can make regarding broader applicability.

Minor comments.

1. The title of the paper needs to be more specific.
2. Abstract, line 6: "the mechanisms behind this phenomenon and the association with human diseases remain unknown" Given that this paper does not address mechanisms, the authors should just focus on the disease relevance.
3. Abstract, line 17: The authors should say "by enhancing transcription" - there is no data that directly ties their observation to transcription initiation.
4. Page 2, paragraph 2: It is very out of place here to have such a lengthy introduction to ASOs here when it is just a tool used in the study to investigate EMATS. The authors should make this section more brief and/or move it to the Results section where they introduce ASOs.
5. Results section, paragraph 1, line 7: This paper does not address translation efficiency. I suggest to not include it in the results section as it is now.
6. Page 4: "We identified human protein coding genes with SEs multiple first exons and selected among those, genes in which the start coordinate of a weak" - the meaning of this sentence is unclear.
7. There are several typos throughout the manuscript. Unfortunately, they are difficult to list here since there are no page/line numbers in the manuscript. I suggest thorough proof-reading to fix them.

Response to Reviewers

We are grateful to the Reviewers for their insightful ideas and valuable suggestions. Their feedback has significantly improved the quality of our manuscript. We have implemented several changes based on their comments. New results include a new pipeline to identify EMATS genes using >17k samples and the identification of tissue-specific EMATS genes, the effect of splicing on SMN2 endogenous gene, the effect of splicing-related genetic mutations on gene expression, newly synthesized RNA as well as RNA polymerase II occupancy and elongation rate measurements following splicing activation, the effect of intron length in this phenomenon, and an exploration of possible RNA binding proteins involved in this regulation. New analyses and experiments are shown in new Figures 1b-c, 2f-g, 3a-c, 5b-d, and 6a-c, as well as in new Supplementary Figures 1a-b, 2a-b, 3a-b, and 5a-c. All new additions to the text are marked in red. Please find below a point-by-point response to the Reviewers' comments.

Reviewer #1 (Remarks to the Author):

This manuscript by Uriostegui-Arcos and colleagues deals with the recently uncovered link that ties alternative splicing with transcriptional activation. Genes where this link exists and the architecture fits (highly included skipped exons located less than 5 kb downstream of a weak alternative promoter) are called EMATS genes. EMATS is an interesting concept that requires careful examination. The study provides a step in this direction and points out many associations between alternative splicing of EMATS genes and stimulation of their transcription. While there may be a subclass of alternatively spliced genes whose structure may be more conducive to transcriptional co-regulation, it is not clear whether architectural parameters will be the most revealing indicators as to how EMATS functions. Overall, I find the study interesting but lacking in clarity and requiring more work: more EMATS genes need to be investigated and more progress is required on molecular mechanisms of action.

1. What is the evidence that transcription is stimulated? In Figure 4 for example, could the increase in steady-state SMN2 mRNA be caused by increased RNA stability. Investigating RNA polymerase occupancy and elongation rate would be needed to add credence to the promoter-linked stimulation of gene expression.

In our original manuscript (Fig. 6c), we demonstrated a significant increase in newly synthesized RNA levels (4sU metabolically labeled) of our splicing reporter following splicing activation with the highest drug concentration under the regulation of all tested promoters. In response to the reviewer's feedback, we have expanded our initial transcriptional analysis by conducting (i) new metabolic labeling experiments using 4sU pulses, (ii) CUT&RUN experiments to analyze RNA polymerase II occupancy, and (iii) RNA polymerase II elongation rate measurements. In all cases, we observed evidence of expression stimulation at the transcriptional level. We observed a significant increase in newly synthesized RNA levels of the splicing reporter following splicing

activation with a small molecule drug, an ASO, and splicing mutations. We observed a significant increase in RNA polymerase II occupancy levels and elongation rate with a small molecule drug and an ASO favoring exon splicing in. To present these new findings, we have included a comprehensive new figure (Fig. 5) in the revised manuscript, which highlights all of these new analyses. We have also referenced this figure in lines 284-321 of the text. Please also find the new figure below.

New Figure 5. a, 30-min 4sU-labelled nascent RNA was extracted from HEK293T-A2 stable cell lines expressing SMN2 splicing reporter under the regulation of five different promoters (CMV, alpha-globin (a-GN), KPTN, Fibronectin wild type (FN-Wt) and Fibronectin mutated (FN-Mut)) (left) or KPTN (right) were treated with Risdaplami (left) or a scramble antisense oligonucleotide (-) or a specific ASO modeled after the Spinraza (+) (right) for 24 h. Expression of SMN2 reporter was evaluated by RT-qPCR. b, 30-min 4sU-labelled nascent RNA was extracted from HEK293T-A2 stable cell lines expressing a splicing reporter in the wild type variant or two mutants that present different inclusion levels of the alternative exon. Expression of the reporter was evaluated by RT-qPCR. c, RNA polymerase occupancy at the promoter region or gene body was evaluated by CUT&RUN using specific antibodies in HEK293T-A2 stable cell lines expressing the SMN2 splicing reporter and treated with Risdaplami (left) or the ASO (right). d, Transcription elongation rate were approximately measured by calculating expression at proximal and distal regions of the SMN2 splicing reporter in HEK293T-A2 stable cell lines following a 30-min metabolically labelling pulse with 4sU.

2. Fig. 4b. Based on the RT-PCR which measures AS, the increase in exon inclusion in SMN2 was certainly not 100-fold as stated given that the PSI went from 0.20 to 0.97 with the highest concentration of risdiplami. If that is the case, I worry that the numbers given about increased SMN2 expression (45-fold and later claims) are also overstatements. While the RT-PCR gels shown are not quantitative and not designed to measure global expression, one would expect the images to display some increase in overall expression, which they are not.

We apologize for the mistake in our previous communication where we incorrectly stated a 100-fold increase in inclusion of SMN2 exon 7. We intended to highlight that a PSI value of 0.97

represents almost complete inclusion (100%). Regrettably, during one of our revisions, this value was mistakenly changed to 100-fold. We have now corrected this error in line 191 of our text. As the reviewer mentioned, RT-PCR gels are not a reliable quantitative method, particularly since we used 30 cycles for these PCRs, which may have led to saturation effects. To address this issue, we repeated the RT-PCR experiments with varying numbers of PCR cycles. Visualization of the gels showed a clear increase in overall SMN2 expression using up to 20 PCR cycles, which avoids saturation effects. We added this experiment in Supplementary Fig. 3a of the manuscript (please also see below).

New Supplementary Figure 3a. HEK293T-A2 stable cell lines expressing SMN2 splicing reporter were treated with Ethanol (-) or 0.1 μM, 1.0 μM, and 10 μM of Risdipiam for 24 h. Inclusion of alternative exon 7 in SMN2 was evaluated by RT-PCR with different cycles. Quantification of densitometric analyses was carried out with Fiji, +Exon/-Exon ratios are shown at the bottom of each lane, and GAPDH is shown as a control.

3. Given that the significant association between skipped exons and gene expression is barely better for EMATS genes (page 8), there is a need to analyze genes that are not in the EMATS category to compare their response with small molecules, ASOs and different promoters. Several examples of EMATS genes would also be required to allow generalization.

Thanks for all these important points. We break this down into different sections below:

3a. Given that the significant association between skipped exons and gene expression is barely better for EMATS genes (page 8)

We agree with the reviewer regarding the small increase in association between skipped exons and gene expression for EMATS genes. We acknowledge that this may have been caused, in part, by the use of highly sensitive computational methods that identified novel alternative exons and the limited representation of tissues and individual variation in our previous analyses. To address this concern, we have revised our approach by focusing only on annotated alternative first and internal exons and excluding the identification of novel exons. Additionally, we have used the entire dataset of >17k samples from the GTEx project to ensure better representation of tissues and individual variation. This new strategy also allowed us to identify tissue-specific EMATS genes, which were not detected in our previous analyses. These new analyses are shown

in new Fig. 1b-c, Fig. 2c-f, new Supplementary Fig. 1a-b, and lines (73 to 93). Below is a summary of the new data.

New Figure 1b,c. Left, number of EMATS genes identified in each human tissue and sub-tissue using data from the GTEx project. Colors represent tissues and grey bars represent tissue-specific EMATS genes, identified only in that tissue. Right, heatmap showing the overlap between EMATS genes identified in each tissue and sub-tissue.

New Figure 2e,f. Left, distribution of the spearman correlation correlations between gene expression and SE PSI across human tissues and samples for non-EMATS genes compared to EMATS genes. Right, distribution of gene expression values in TPM for the NSFM gene in brain amygdala samples binned by quartiles of inclusion in PSI values for the SE contributing to the EMATS structure.

3b. ...there is a need to analyze genes that are not in the EMATS category to compare their response with small molecules, ASOs and different promoters.

In order to analyze genes that are not in the EMATS category, we opted to modify our system, adding introns of different lengths to increase the distance between the skipped exon and the transcription start site (TSS) until the reporter would not be considered to have an "EMATS"

structure any more. We observed a gradual decrease in the effect on gene expression triggered by the same increase in inclusion of the skipped exon when we used introns of greater length. Further, we analyzed the endogenous SMN2 gene which does not support a coding transcript with EMATS structure as the primary ORF requires initiation at a TSS > 10kb upstream from the gene's skipped exon. We did observe an increase in expression of the endogenous SMN2 gene following splicing activation, but it was significantly lower compared with the reporter where the TSS is located less than 10kbs upstream of the alternative exon 7. We added new figure panels to the manuscript with these results (Fig. 3a-c, Fig. 6a, and Supplementary Fig. 5a-b) and a new section of the text in lines 185-195, and 328-337. Please find below a summary of the main results.

New Figure 6a. HEK293T-A2 stable cell lines expressing SMN2 splicing reporter under the regulation of Fibronectin (FN) promoter and three different intronic sequences were treated with Risdipiam (left) or a scramble antisense oligonucleotide (-) or a specific ASO modeled after the Spinraza (+) for 24 h (right). Expression of the SMN2 reporter was evaluated by RT-qPCR.

New Figure 3a-c. Top, diagram of SMN2 gene highlighting alternative exon 7 and the binding sites of Risdiplam and the ASO modeled after the Spinraza. Wild type HEK293 cells were treated with Ethanol (-) or 0.1 μ M, 1.0 μ M, and 10 μ M, of Risdiplam for 24 h (left) or a scramble antisense oligonucleotide (0) or different concentrations of a specific ASO modeled after the Spinraza for 24 h (right). Inclusion of alternative exon 7 in SMN2 was evaluated by RT-PCR. Quantification of densitometric analyses was carried out with Fiji, +Exon/-Exon ratios are shown at the bottom of each lane. SMN2 expression was evaluated by RT-qPCR.

3c. Several examples of EMATS genes would also be required to allow generalization.

It is not possible to test this in other EMATS genes since we would need an existing and available technology (other ASOs or small molecules) to activate inclusion of alternative exons in other EMATS genes, and we are not aware of that. We are currently trying to develop this technology for disease-related genes, but it will of course take many years of work (like it was necessary to develop Risdiplam and Spinraza). Please notice that we already showed that downregulation of several endogenous EMATS genes can be achieved through ASOs that block splice sites of appropriately located alternative exons in Fiszbein et al., 2019.

Based on the reviewer suggestion and to contribute to the generalization of these effects, we were able to mutate the splice sites of another EMATS-like construct to achieve lower and higher levels of inclusion of the alternatively spliced internal exon. As expected, weak splice sites favored exon exclusion and triggered a downregulation of overall expression of the transcript, while strong splice sites favored exon inclusion and triggered an upregulation of transcript expression. We added this experiment in Fig. 2g of the manuscript (please also see below) and highlighted it in lines 146-152 of the text.

New Figure 2g. Diagram of splicing reporter indicating the splice site scores of the alternative exon in the wild type construct and two mutants that present different PSI values by RT-PCR. Quantification of densitometry analyses was carried out with Fiji, +Exon/-Exon ratios are shown at the bottom of each lane. h, expression of three splicing reporters -wild type, strong and weak splice sites (ss)- integrated in cells was evaluated by RT-qPCR after.

4. How does the impact of risdiplam on SMN2 splicing and gene expression compare with that of the ASO? i.e., does a similar change in PSI yield a similar stimulation of mRNA expression?

To be able to compare the impact of Risdiplam with that of the ASO we expanded our experiment using several ASO concentrations. At low concentrations, the ASO showed weaker effects on gene expression compared to Risdiplam, but at high concentrations, the ASO promoted a larger change in expression through a similar splicing change. This is shown in the new Fig. 3d and new Supplementary Fig. 3b; please also see below.

New Figure 3d and Supplementary Figure 3b. Left, stable HEK293T-A2 cells expressing SMN2 splicing reporter were treated with a scramble antisense oligonucleotide (-) or different concentrations of a specific ASO modeled after the Spinraza (+) for 24 h (right). Inclusion of alternative exon 7 in SMN2 was evaluated by RT-PCR and SMN2 expression was evaluated by RT-qPCR. Right, change in expression compared to change in splicing for cells expression SMN2 reporter following treatment with Risdiplam or a specific ASO modeled after the Spinraza for 24 h.

5. The claim that genetic variants reprogram splicing and gene expression should be investigated more thoroughly using reporter minigenes.

We agree with the reviewer. Unfortunately, our cloning strategy was not successful, and we were not able to investigate the case of FHL1 gene with reporter minigenes as intended. Based on both reviewers' feedback, and since it was not central to the story, we decided to remove that section of the manuscript. We no longer include FHL1 and MBT7 examples. However, we were able to investigate the effect of genetic mutations that affect splicing in gene expression using reporter minigenes and added it to New Fig. 2g. This is shown in Response to Reviewer #1 comment 3c.

6. An alternative and possibly more mundane explanation for EMATS may be that EMATS genes are in general longer producing longer transcripts. Such long transcripts are likely being bound by more factors and have more introns. Thus, they may have more time to stochastically move to areas of the nucleus where splicing and transcription factors exist at higher concentrations. Likewise, improving spliceosome interactions of alternatively spliced transcripts that are still in association with their DNA may increase the residency time of such transcripts in areas rich in transcription factors, hence explaining stimulation.

This is a very interesting idea. While EMATS genes are in general longer, the usage of exons in an EMATS structure produce similar-sized or smaller transcripts (please see Supplementary Fig. 1d) which would stochastically recruit less factors.

Other comments

1. Not quite sure what Supplementary Table 2 is, and no description is offered. May be indicate the names of genes along with their coordinates. Supplementary Table 2 was a list of EMATS genes with novel exons that were previously not annotated as first or internal exons, but discovered by our pipeline. We no longer allow for discovery of new exons (please see response to Reviewer #1 comment 3a), so we've deleted that table. Supplementary Table 2 now refers to tissue-specific EMATS genes.

2. Intro: What is a "highly included skipped exon"? A highly included skipped exon is an exon with a median percent splice-in (PSI) value greater than the dataset-wide median of all skipped exon PSI values. The definition is highlighted in page 20 lines 500-502. We added a brief description in the introduction as well.

3. EMATS must be rare enough to explain why transcription regulation and alternative splicing regulation have always been considered distinct layers of regulation (i.e., genes displaying tissue-specific AS regulation are not the one displaying tissue-specific transcriptional regulation)? Does its actual prevalence match this prediction? We have previously showed that alternative splicing of skipped exons is associated with transcriptional activation of nearby upstream transcription start sites across tissues and species (Fiszbein et al., 2019). Other studies also found a positive association between AS and promoter usage. We don't think EMATS is rare, but it is limited to genes with an efficiently spliced alternative exon located downstream of and nearby a weak promoter.

4. Fig. 2c, what are the two sets of numbers in FLH1B? When referring to patients with genetic variants in FLH1, the difference in the expression of each variant relative to normal should be clearly stated to clarify what exactly the mutations elicit. Similarly, for MBOAT7, it is not clear from the text what the mutation promotes. The two sets of numbers correspond to the splice site scores measured by MaxEnt score for the two different sets of mutations A → T/G and G → T/A. Since we were not able to validate the expression of each variant, we removed that section of the manuscript.

5. Only the best cases are shown in Fig. 3c. What is the range of R² for all EMATS investigated? Having expanded our sample size to the entire GTEx RNA-seq dataset, we observed increased non-linearity between skipped exon usage and gene expression and, therefore, opted to report just the Spearman rank correlation in figure 2f (previously 3c). Still, we performed a linear least-squares regression with each skipped exon and report the R² distribution for EMATS genes in Supplementary Figure 2a.

6. Page numbering would be useful. We added page and line numbers.
7. Fig. S1a and S1b are reversed relative to the text. Fixed
8. Why isn't the 5'ss of SE not indicated as having a significant number of variations when the 3'ss is? (see Fig. 2b). We redid all of these analyses using our new set of EMATS genes based on the expanded sample size. We found the majority of variants located within the EMATS AFE and SEs and a significant number in both splice sites.
9. For FHL1, does the "downregulation of the second TSS by EMATS" the same as a downregulation of FHL1B? Please indicate what wAFE and sSE stand for in the figure legend? Yes, the second TSS expresses the FHL1B isoform. wAFE correspond to weak AFE and sSE to strong SE. We removed that section of the manuscript.
10. Page 8: reference needed for association of new exons with increased expression. Fixed
11. SARS-Covid2 is not a term (Suppl Fig. 2). Replaced by SARS-CoV-2

Reviewer #2 (Remarks to the Author):

Splicing-dependent transcriptional activation

SUMMARY: In this study, Uriostegui-Arcos et al. systematically catalog genes that can undergo exon-mediated activation of transcription starts (EMATS) and investigate their potential to be transcriptionally activated by splicing modulation. They computationally identify two genes where pathogenic variants might disrupt protein domains via EMATS. Using a minigene reporter of SMN2 splicing, they show that both ASOs and small molecules that promote inclusion of exon 7 boost the expression of the reporter. These results are promising and suggest the possibility of EMATS as a mechanism underlying disease, as well as its potential for combating haploinsufficiency based genetic diseases. However, there are several limitations to this study that reduce my enthusiasm.

The authors make claims throughout the manuscript that overreach what their data demonstrates. For example, they claim that their Fig 2 "provide(s) evidence for protein structure disruption in specific EMATS genes caused by genetic variants". From my understanding, Fig 2 shows how specific genetic variants in these predicted EMATS genes could theoretically impact splicing and disrupt protein domains. However, the authors state that "Patients with these genetic variants express a wild type FHL1C, a mutated version of FHL1B which disrupts a highly conserved double zinc finger domain called LIM domain33, and reduced levels of a mutated version of FHL1A by EMATS". Isn't this a prediction based on the splicing score? Have the authors measured the observed splicing pattern for these genes in patients with mutations? If

they do not have such data (and it is not clear from the description that they do), they should not make a strong claim as to the "expression" of various splice isoforms in patients. They also do not show any experimental data to conclusively connect this phenomenon to EMATS.

We agree with the reviewer. That section of the manuscript was predictive and our language about FHL1 isoforms' expression in patients was too strong. Since this was not central to the story, we decided to remove those examples from the manuscript and do not make any claims until we are able to conclusively connect this phenomenon to EMATS. However, we were able to investigate the effect of genetic mutations that affect splicing on gene expression using reporter minigenes. We mutated the splice sites of an EMATS-like construct to achieve lower and higher levels of inclusion of the alternatively spliced internal exon. As expected, weak splice sites favored exon exclusion and triggered a downregulation of overall expression of the transcript, while strong splice sites favored exon inclusion and triggered an upregulation of transcript expression. We added this experiment in Fig. 2g of the manuscript (please also see below) and highlighted it in lines 146-152 of the text.

New Figure 2g. Diagram of splicing reporter indicating the splice site scores of the alternative exon in the wild type construct and two mutants that present different PSI values by RT-PCR. Quantification of densitometry analyses was carried out with Fiji, +Exon/-Exon ratios are shown at the bottom of each lane. h, expression of three splicing reporters -wild type, strong and weak splice sites (ss)- integrated in cells was evaluated by RT-qPCR after.

Another major weakness of this manuscript is that the authors do not address how much of the observed upregulation of gene expression is due to EMATS vs other phenomenon including enhanced stability or transport. While they acknowledge this limitation in one part of the manuscript, they routinely conflate increased mRNA level with enhanced transcription initiation due to EMATS throughout the paper. The only piece of data to suggest transcriptional upregulation is in the last figure where they measure nascent RNA level of the SMN2 reporter. The authors should more carefully account for the relative contribution of transcription induction through additional metabolic labeling experiments.

We thank the reviewer for this important point. In response to the reviewers', we have expanded our initial transcriptional analysis by conducting (i) new metabolic labeling experiments using 4sU pulses, (ii) CUT&RUN experiments to analyze RNA polymerase II occupancy, and (iii) RNA polymerase II elongation rate measurements. In all cases, we observed evidence of expression stimulation at the transcriptional level. We observed a significant increase in newly synthesized RNA levels of the splicing reporter following splicing activation with a small molecule drug, an ASO, and splicing mutations. We observed a significant increase in RNA polymerase II occupancy levels and elongation rate with the small molecule and an ASO favoring splicing in. To present these new findings, we have included a comprehensive new figure (Fig. 5) in the revised manuscript, which highlights all of these new analyses. We have also referenced this figure in lines 284-321 of the text. We also re-worded some parts of the manuscript to not conflate mRNA levels with transcriptional upregulation. Please also find the new figure below.

New Figure 5. EMATS activates transcription initiation and elongation. **a**, 30-min 4sU-labelled nascent RNA was extracted from HEK293T-A2 stable cell lines expressing SMN2 splicing reporter under the regulation of five different promoters (CMV, alpha-globin (α -GN), KPTN, Fibronectin wild type (FN-Wt) and Fibronectin mutated (FN-Mut)) (left) or KPTN (right) were treated with Risdiplam (left) or a scramble antisense oligonucleotide (-) or a specific ASO modeled after the Spinraza (+) (right) for 24 h. Expression of SMN2 reporter was evaluated by RT-qPCR. **b**, 30-min 4sU-labelled nascent RNA was extracted from HEK293T-A2 stable cell lines expressing a splicing reporter in the wild type variant or two mutants that present different inclusion levels of the alternative exon. Expression of the reporter was evaluated by RT-qPCR. **c**, RNA polymerase occupancy at the promoter region or gene body was evaluated by CUT&RUN using specific antibodies in HEK293T-A2 stable cell lines expressing the SMN2 splicing reporter and treated with Risdiplam (left) or the ASO (right). **d**, Transcription elongation rate were approximately measured by calculating expression at proximal and distal regions of the SMN2 splicing reporter in HEK293T-A2 stable cell lines following a 30-min metabolically labelling pulse with 4sU.

Finally, the authors need to demonstrate this phenomenon in an endogenous EMATS gene. All of their conclusions are based on an integrated minigene reporter and it limits the scope of the claims they can make regarding broader applicability.

Based on the Reviewer suggestion, we tested our system in the endogenous SMN2 gene. Notably, both the ASO and the small molecule upregulated expression of the endogenous SMN2 gene. These new sets of experiments were included in new Figure 3a; please also see below.

Please note that it is not possible to test this in other endogenous EMATS genes since we would need an existing and available technology (other ASOs or small molecules) to activate inclusion of alternative exons in other EMATS genes, and we are not aware of that. We are currently trying to develop this technology for disease-related genes, but it will take many years of work (as was necessary to develop Risdiplam and Spinraza). Please note that we previously showed that downregulation of several endogenous EMATS genes can be achieved through ASOs that block splice sites of appropriately located alternative exons in Fiszbein et al., 2019.

New Figure 3a-c. Top, diagram of SMN2 gene highlighting alternative exon 7 and the binding sites of Risdiplam and the ASO modeled after the Spinraza. Wild type HEK293 cells were treated with Ethanol (-) or 0.1 μM, 1.0 μM, and 10 μM, of Risdiplam for 24 h (left) or a scramble antisense oligonucleotide (0) or different concentrations of a specific ASO modeled after the Spinraza for 24 h (right). Inclusion of alternative exon 7 in SMN2 was evaluated by RT-PCR. Quantification of densitometric analyses was carried out with Fiji, +Exon/-Exon ratios are shown at the bottom of each lane. SMN2 expression was evaluated by RT-qPCR.

Minor comments.

1. The title of the paper needs to be more specific. We changed the title to “Splicing activates transcription from weak promoters upstream of alternative exons”, but we still prefer our

previous shorter and simpler title. We ask the reviewer to please let us know if they think this change is necessary.

2. Abstract, line 6: "the mechanisms behind this phenomenon and the association with human diseases remain unknown" Given that this paper does not address mechanisms, the authors should just focus on the disease relevance. We have updated the manuscript by removing "the mechanisms behind..." from the abstract. However, we have included a new exploration of a potential mechanism underlying this phenomenon. Specifically, we conducted an analysis of the correlation between gene expression and skipped exon inclusion using RNA-seq data from a recent ENCODE project, which involved the knock-down of over 250 RNA binding proteins (RBPs). Our analysis revealed a positive association between global changes in gene expression and exon skipping, consistent with our previous findings across human tissues. Importantly, this positive association was lost following knock-down of several RBPs. To identify the RBPs that contribute most strongly to this association, we selected those with the largest residuals from the linear regression of splicing and gene expression changes. We then built a protein-protein interaction network for these RBPs and found that they primarily interact with transcription factors. These results suggest a possible local interaction model, in which RBPs recruited during splicing interact with transcription factors to jointly impact the expression and splicing of the same genes. These new results are shown in new Fig. 6b–d, please also see below.

Figure 6. (a) HEK293T-A2 stable cell lines expressing SMN2 splicing reporter under the regulation of Fibronectin (FN) promoter and three different intronic sequences were treated with Risdipiam (left) or a scramble antisense

oligonucleotide (-) or a specific ASO (+) for 24 h (right). Expression of the SMN2 reporter was evaluated by RT-qPCR. The scatter plot with bars represents the mean \pm standard error, and individual dots represent individual data points. Bars with the same letter indicate no significant difference between means, as determined by one-way ANOVA (Tukey's HSD test, $p < 0.0001$). (b), correlation between the fraction of differentially upregulated (left) and downregulated (right) genes and the fraction of differentially upregulated or downregulated skipped exons following RBP knock-down. RBPs with largest residuals are highlighted in red. (c), PPI network for the 17 unique RBPs from (b) with the largest residuals, colored by Gene Ontology category (5 RBPs we not connected to the network and excluded from the visualization). PPI data are from the STRING database. (d), model showing that splicing can activate transcription initiation depending on three main factors: the strength of the promoter, the inclusion of the skipped exon, and the genomic distance between them. Splicing factors (SF) recruited to the splicing event interact with transcription factors (TF) increasing the local concentration of TFs and inducing transcription from proximal promoters which, in turn, favors inclusion of skipped exons.

3. Abstract, line 17: The authors should say "by enhancing transcription" - there is no data that directly ties their observation to transcription initiation. *Fixed - although the new CUT&RUN experiments do suggest enhancement of transcription initiation by showing an increase in RNA polymerase occupancy at the promoter region.*

4. Page 2, paragraph 2: It is very out of place here to have such a lengthy introduction to ASOs here when it is just a tool used in the study to investigate EMATS. The authors should make this section more brief and/or move it to the Results section where they introduce ASOs. *We made this section briefer which resulted in removing some citations.*

5. Results section, paragraph 1, line 7: This paper does not address translation efficiency. I suggest to not include it in the results section as it is now. *We removed the reference to previous results on translation efficiency.*

6. Page 4: "We identified human protein coding genes with SEs multiple first exons and selected among those, genes in which the start coordinate of a weak" – the meaning of this sentence is unclear. *We changed it to "To narrow down our focus, we first identified human protein-coding genes that possessed both SEs and multiple first exons. We then further refined our selection by choosing genes in which a weak alternative first exon (AFE) was located within 5 kb upstream of one or more highly included SE's 5' coordinate".*

7. There are several typos throughout the manuscript. Unfortunately, they are difficult to list here since there are no page/line numbers in the manuscript. I suggest thorough proof-reading to fix them. *We thoroughly proof-read the manuscript and added page and line numbers. We hope to not have missed any typos this time.*

REVIEWERS' COMMENTS

Reviewer #1 (Remarks to the Author):

I find this resubmission dramatically improved. The authors have clearly paid attention to all comments and have reconstructed portions of the manuscript to make it much clearer, and to the point. Other than typos that should be spotted during formatting I do not have any serious criticisms on this version which I would recommend for publication.

Reviewer #2 (Remarks to the Author):

The authors have adequately addressed my concerns through new experiments/analysis and by scaling back some of their claims. I support the publication of the revised manuscript. It is an important advance for the field.

I would prefer for the authors to keep their more specific revised title for the manuscript.